# Feasibility and performance of the fecal immunochemical test (FIT) for average-risk colorectal cancer screening in Nigeria

Gregory C. Knapp[1]¤*, Olusegun Alatise[2], Bolatito Olopade[3], Marguerite Samson[4], Olalekan Olasehinde[2], Funmilola Wuraola[2], Oluwole O. Odujoko[5], Akinwunmi O. Komolafe[5], Olujide O. Arije[6], Philip E. Castle[7], J. Joshua Smith[1], Martin R. Weiser[1], T. Peter Kingham[1]

1 Department of Surgery, Memorial Sloan Kettering Cancer Center, New York, New York, United States of America, 2 Department of Surgery, College of Health Sciences, Obafemi Awolowo University, Ile-Ife, Nigeria, 3 Department of Medical Microbiology and Parasitology, College of Health Sciences, Obafemi Awolowo University, Ile-Ife, Nigeria, 4 Department of Epidemiology and Biostatistics, Memorial Sloan Kettering Cancer Center, New York, New York, United States of America, 5 Department of Morbid Anatomy and Forensic Medicine, College of Health Sciences, Obafemi Awolowo University, Ile-Ife, Nigeria, 6 Institute of Public Health, Obafemi Awolowo University, Ile-Ife, Nigeria, 7 Department of Epidemiology and Public Health, Albert Einstein College of Medicine, Bronx, New York, United States of America

¤ Current address: Division of General Surgery, Department of Surgery, Dalhousie University, Halifax, Nova Scotia, Canada
* knappg@pm.me

**Data Availability Statement:** All relevant data are within the manuscript and its Supporting Information files.

## Abstract

### Introduction

There is a paucity of prospective data on the performance of the fecal immunochemical test (FIT) for colorectal cancer (CRC) screening in sub-Saharan Africa. The aim of this exploratory analysis was to evaluate the feasibility and performance of FIT in Nigeria.

### Methods

This was a prospective, single-arm study. A convenience sample of asymptomatic, average-risk individuals between 40–75 years of age were enrolled at Obafemi Awolowo University Teaching Hospital. Study participants returned in 48 hours with a specimen for ova and parasite (O&P) and qualitative FIT (50ug/g) testing. Participants with a positive FIT had follow-up colonoscopy and those with intestinal parasites were provided treatment.

### Results

Between May-June 2019, 379 individuals enrolled with a median age of 51 years (IQR 46–58). In total, 87.6% (n = 332) returned for FIT testing. FIT positivity was 20.5% (95% CI = 16.3%-25.2%). Sixty-one (89.7%) of participants with a positive FIT had a follow-up colonoscopy (n = 61), of whom 9.8% (95%CI:3.7–20.2%) had an adenoma and 4.9% (95%CI:1.0–13.7%) had advanced adenomas. Presence of intestinal parasites was inversely related to FIT positivity (6.5% with vs. 21.1% without parasites, p = 0.05). Eighty-two percent of

**Funding:** The authors received no specific funding for this work.

**Competing interests:** The authors have declared that no competing interests exist.

participants found the FIT easy to use and 100% would recommend the test to eligible family or friends if available.

## Conclusions

Asymptomatic, FIT-based CRC screening was feasible and well tolerated in this exploratory analysis. However, the high FIT positivity and low positive predictive value for advanced neoplasia raises concerns about its practicality and cost effectiveness in a low-resource setting such as Nigeria.

## Introduction

The incidence of colorectal cancer (CRC) in Nigeria is increasing [1, 2]. To address the rising disease burden, the National Cancer Control Plan (NCCP 2018–2022) identifies CRC screening and early detection as a priority for health system investment and capacity building [3]. Several organizations, including the American Society of Clinical Oncology, have published resource-stratified guidelines to guide the implementation of CRC screening and early detection policies [4–6]. For countries such as Nigeria, with limited endoscopy capacity, the fecal immunochemical test (FIT) for occult blood is advocated as the preferred screening modality. This is based on a robust body of peer-reviewed literature that supports the efficacy of the test in high-income countries (HIC) [7–10]. Unfortunately, the performance of stool-based CRC screening has not been evaluated in low-income countries like Nigeria. The cultural institutions, socioeconomic development (e.g. health insurance, flush toilets) and basic infrastructure (e.g. health care access travel time) are also markedly different in Nigeria from corresponding environments of Europe and North America where FIT was validated.

The pathogenesis of CRC has not been as thoroughly characterized in West Africa as it has in HICs due to a lack of widespread screening and molecular testing. In those settings where high-quality endoscopy is available, there appears to be a low burden of traditional adenomas, with adenoma detection rates (ADR) of <10% in adult series [11–15]. These findings support preliminary data suggesting a unique molecular profile (i.e. 23% with microsatellite instability) of CRC in West Africa [16–19]. This raises further uncertainty regarding the translatability of FIT performance from HICs to Nigeria.

The aim of this study was to prospectively explore the feasibility of stool-based CRC screening in Nigeria. The primary endpoint was the positive predictive value of FIT for advanced neoplasia detection, with endoscopic follow-up, in asymptomatic Nigerian adults with concomitant intestinal parasite testing.

## Methods

### Study population

Between May–June 2019, a convenience sample of asymptomatic, average-risk individuals between 40–75 years of age were sought from the city of Ile-Ife, Nigeria and the surrounding rural villages (e.g. Ijebu Jesa). The study was promoted using print, radio and social media. Volunteers were excluded from enrollment if they had: a personal history of CRC; a first degree relative with CRC; self-reported rectal bleeding within the last 6 months; or a colonoscopy within the last 5 years. Eligibility was congruent with current CRC screening guidelines endorsed by the Society of Gastroenterology and Hepatology in Nigeria as well as programs of

organized FIT-based screening in HICs [20]. The lower age threshold for inclusion (i.e. 40) in this study reflects the lower median age (i.e. 50–52) of CRC diagnosis in Nigeria [19]. Female participants were asked to wait three days from the end of menses before providing a stool sample.

## Specimen collection and processing

Participants were administered an enrollment questionnaire to elicit data on sociodemographic variables and recent medical history. All participants were asked to return in 48 hours with a fresh stool specimen for parasite testing as well as a completed FIT kit for analysis. Teaching on proper technique for dry, uncontaminated stool capture was individually administered in either English or Yoruba (i.e. local language). Returning participants were administered a questionnaire to elicit perceptions of stool-based CRC screening while the FIT was processed. Participants with a positive FIT were offered a follow-up colonoscopy and those with intestinal parasites were provided appropriate treatment.

## Fecal immunochemical test

The study was designed to explore the performance of FIT, which has largely replaced guaiac fecal occult blood testing (gFOBT) as the preferred modality for population-based CRC screening in HICs [7]. FIT is associated with greater participant compliance and a higher positive predictive value (PPV) for advanced neoplasia compared to gFOBT [8]. A lack of dietary restrictions, single stool sample (as opposed to two or three for gFOBT) and superior cost-effectiveness in population-based programs, were important considerations in choosing FIT over gFOBT [9].

Each participant was given a Medline iFOB (Medline Industries Inc. Northfield Il., Lot nom. 768L11) collection tube. The Medline iFOB is a Clinical Laboratory Improvement Act (CLIA)-waived, qualitative FIT product with a manufacturer-set, lower limit of hemoglobin detection of 50 g/g. Stool was exposed to ambient temperature for no longer than 48 hours between evacuation and processing. Each specimen was processed as per the manufacturer's instructions, including verification of activated internal control. The result for each test was interpreted by two members of the research team.

## Parasitology

Participants provided a separate stool sample for parasite testing. Saline and iodine mounts of direct stool smears were prepared for examination under light microscopy at 10x and 40x power. The formalin-ether sedimentation technique was used to concentrate the specimens. Modified Ziehl Neelson staining of the concentrated stool specimens was employed to detect cryptosporidium oocysts and other intestinal coccidian parasites, such as Isospora sp., and Cyclospora sp. All individuals infected with intestinal parasites were contacted after final results were available and provided treatment.

## Follow-up colonoscopy

Individuals with a positive FIT screen were counselled and provided with a two-day, polyethylene glycol 3350-based bowel preparation and colonoscopy follow-up appointment. Colonoscopy was performed by two high-volume endoscopists at Obafemi Awolowo University Teaching Hospital (OAUTH) with the aid of conscious sedation. Data from the colonoscopy, including quality of bowel preparation, cecal intubation, anatomic location of gross findings, and withdrawal time were prospectively collected. The histopathology report was collected at

the time of outpatient follow-up. Advanced adenomas were defined as traditional adenomas with tubulovillous (25% villi) or villous architecture (75% villi), high-grade dysplasia, or size 1 cm [21, 22]. Advanced neoplasia included both advanced adenomas and adenocarcinoma. Localization of endoscopic findings was described as corresponding to the right-colon (i.e. proximal to the splenic flexure), left-colon (i.e. rectum to splenic flexure), and rectum.

## Statistical analysis

Descriptive statistics were performed on the sociodemographic, past medical history, and cancer perceptions variables. Bivariate analysis was performed to test for factors associated with FIT positivity. Binomial 95% confidence intervals were calculated for FIT positivity and the positive predictive value of a positive FIT for adenoma and advanced adenoma. A two-tailed *p*-value level of 0.05 was used to denote significance. Based on the results of our previously published analysis of FIT performance, we calculated a target enrollment of 264 participants to have 80% power to detect a difference in FIT positivity between those with parasitic infection vs. those without with an alpha of 0.05 [23]. Personal and household income were converted from the local currency (Naira) to dollars (USD) using the Central Bank of Nigeria's conversion rate on May 2, 2019 (N = 305.95).

## Ethics

Institutional research board approval from Obafemi Awolowo University (OAU) in Ile-Ife, Nigeria was granted for this study, including the follow-up endoscopy for a positive FIT result. This study is registered with clinicaltrial.gov (identifier: NCT03473795). Recruitment was voluntary and the cost of the interventions were provided free of charge. Informed written consent was obtained in either English or Yoruba from each participant.

## Results

We enrolled 379 participants, with a median age and monthly income of 51 years (IQR 46–58 years) and $653.7 (IQR $274.6-$1307.4), respectively. The majority of participants were female (70.7%, 268/379) and 64.7% (246/379) had at least a university-level education. The complete sociodemographic profile of the cohort is presented in Table 1. Forty percent (153/379) of participants had previously undergone some form of cancer screening. However, only 6.6% (25/371) had previously discussed CRC screening with a physician. Of the 379 who enrolled, 332 (87.6%) returned their FIT kits for processing (Fig 1). Twenty-eight of the 332 (8.4%) FIT kits were collected improperly (e.g. too much stool) but were successfully processed after additional specimen dilution with buffer solution.

When participants returned for FIT processing, perceptions of stool-based CRC screening were elicited. One hundred percent of participants indicated that they would participate in regular CRC screening if it was free, and would recommend FIT to family or friends if available. Eighty-two percent (273/332) of participants strongly agreed with the statement: "the FIT kit (CRC screening test) was easy to use". The most common issue captured by the participants on qualitative assessment was the diminutive size of the collection tube.

Overall FIT positivity was 20.5% (68/332) (95%CI: 16.3–25.2%). The likelihood of a positive test was similar between participants aged 50 (18.5%, 27/146) and > 50 (22%, 41/186, p = 0.43) years of age. Sixty-six percent (45/68) of the positive results were in women and 36.8% (25/68) in participants aged 40–49 years. On bivariate analysis, gender (male 23% vs. female 19.5%, p = 0.47), education level (primary 13.3%, secondary 16.3%, vocational 18.8%, university 23.3%, graduate 19.3%, p = 0.37), and income (p = 0.64) were not associated with FIT positivity (Table 2). Self-reported hypertension (p = 0.26) and peptic ulcer disease (p = 0.73) were also

**Table 1. Sociodemographic and past medical history characteristics.**

| Characteristic | n = 379 | % |
|---|---:|---:|
| **Age (years)** | | |
| 40–49 | 157 | 41.4 |
| 50–59 | 143 | 37.7 |
| 60–69 | 62 | 16.4 |
| 70 | 17 | 4.5 |
| **Gender** | | |
| Male | 110 | 29.1 |
| Female | 268 | 70.9 |
| **Monthly house income** | | |
| <N49,999 ($163) | 36 | 14.6 |
| N50-99,999 ($163–327) | 30 | 12.1 |
| >N100,000 ($327) | 181 | 73.3 |
| **Level of education** | | |
| No formal education | 6 | 1.6 |
| Primary | 38 | 10.0 |
| Secondary | 49 | 12.9 |
| Vocational | 40 | 10.6 |
| Undergraduate | 146 | 38.5 |
| Graduate | 100 | 26.4 |
| **Comorbidities** | | |
| Diabetes mellitus | 26 | 6.9 |
| Hypertension | 98 | 25.9 |
| Heart burn / dyspepsia | 76 | 20.2 |
| **Medication use** | | |
| SA or NSAIDS | 105 | 27.8 |
| **Constitutional symptoms** | | |
| Change in bowel habits | 25 | 6.6 |
| New weight loss* | 14 | 3.7 |
| New fatigue | 33 | 8.7 |

*unintentional

not associated with FIT positivity. Twenty-seven percent (26.5%, 88/332) of the cohort reported taking non-steroidal anti-inflammatory drugs (NSAID) or acetylsalicylic acid (ASA) within seven days of FIT testing and neither were associated with a positive result (p = 0.99). Seven percent of participants endorsed new onset fatigue (7.8%, 26/332) within six months of enrollment, while 3.6% (12/332) and 6.3% (21/332) reported weight loss and change in bowel habits, respectively. None of these variables were associated with FIT positivity (S1 Table).

Twelve percent (11.5%, 32/279) of participants had stool specimens positive for intestinal parasites. The most common pathogen was *Entamoeba coli* (53.1%), followed by *Entamoeba histolytica* (31.3%) and *Cryptosporidium* (18.8%). In total, 268 participants had results for both FIT and parasite testing. Participants with asymptomatic intestinal parasites were less likely to have a positive FIT than participants without parasites (6.5% vs. 21.1%, p = 0.05, Table 3).

In total, 68 participants with a positive FIT were offered a follow-up colonoscopy. Seven patients (10.3%) did not have colonoscopy either because they declined (n = 6) or was lost to follow-up (n = 1). The majority of participants (81.8%, 50/61) had an excellent quality bowel preparation, which was associated with a 95.1% (58/61) cecal intubation rate and mean

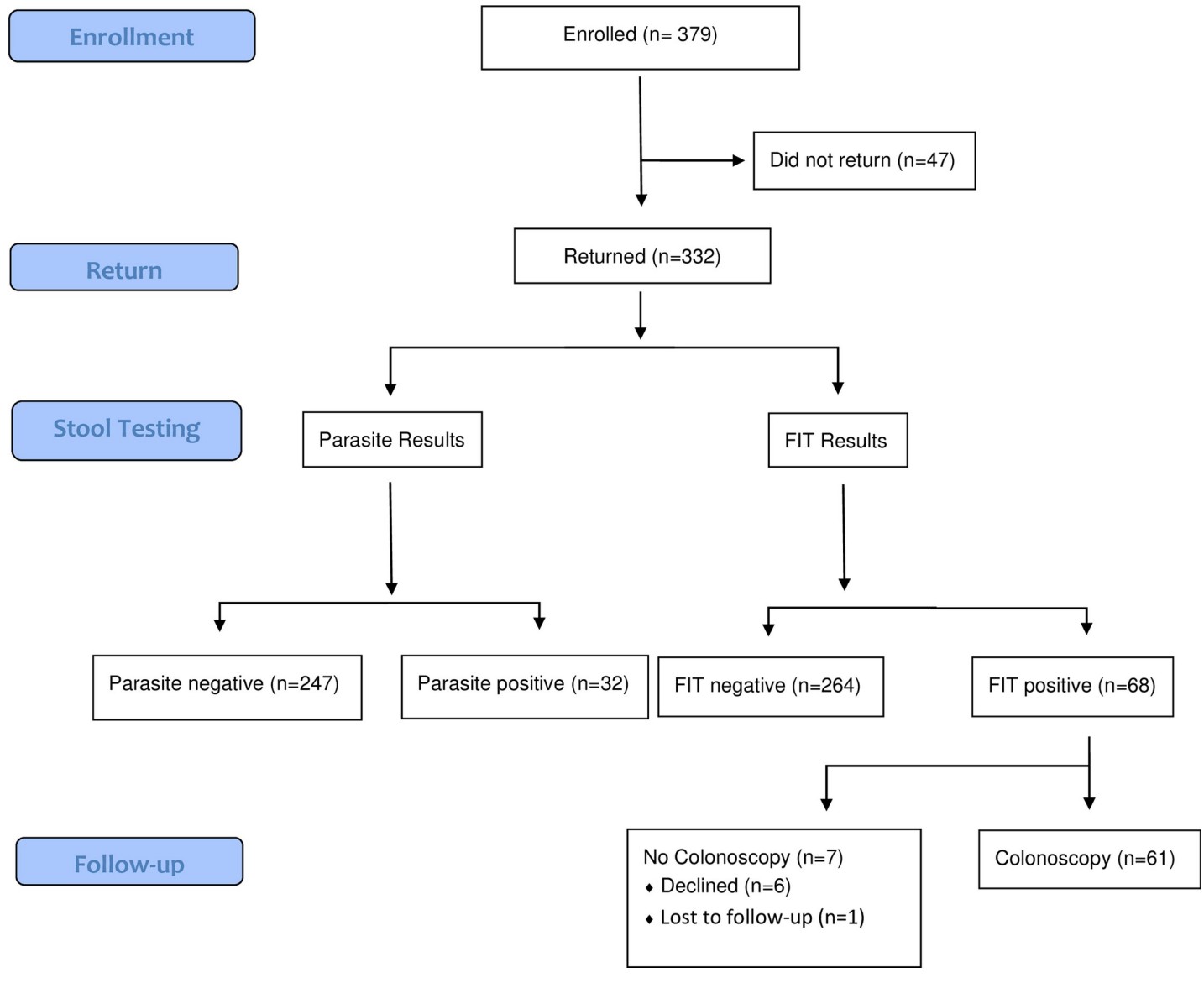

**Fig 1. CONSORT flow diagram.**

endoscopy withdrawal time of 8.5 min for negative endoscopies. There were no reported complications, including post-polypectomy bleeding, perforation, or 30-day re-admission.

Of the 61 FIT-positive patients who underwent colonoscopy, polyps were detected in 23.0% (14/61, 95% CI:13.2–35.5%) of participants and histopathology was available for 85.7% (12/14). Overall, 9.8% (6/61; 95% CI: 3.7–20.2%) had adenomas. Two inflammatory polyps were removed, but no serrated adenomas (i.e. hyperplastic or sessile serrated) or adenocarcinomas were detected (0.0%, 95%CI: 0.0–5.9%) (Table 4). On bivariate analysis, sociodemographic variables (Table 5), comorbidities (HTN, diabetes), ASA or NSAID use within seven days, and recent change in constitutional symptoms (new fatigue, weight loss, and change in bowel habits) were not associated with adenoma detection (Table 6). However, cigarette smoking (50.5%), history of HTN (15.8%), and graduate education (17.6%) demonstrated higher adenoma detection rates.

**Table 2. Bivariate analysis for sociodemographic variables associated with FIT result.**

| Covariate | FIT result | | | | p value* |
| --- | --- | --- | --- | --- | --- |
| | Negative | | Positive | | |
| | *n* | % | *n* | % | |
| **Age** | | | | | |
| 40–49 yrs | 106 | 80.9 | 25 | 19.1 | 0.91 |
| 50–59 yrs | 102 | 79.1 | 27 | 20.9 | |
| 60–69 yrs | 45 | 78.9 | 12 | 21.1 | |
| 70 yrs | 11 | 73.3 | 4 | 26.7 | |
| **Monthly household income** | | | | | |
| <N49,999 ($163) | 26 | 83.9 | 5 | 16.1 | 0.64 |
| N50-99,999 ($163–327) | 22 | 84.6 | 4 | 15.4 | |
| >N100,000 ($327) | 127 | 78.4 | 35 | 21.6 | |
| **Gender** | | | | | |
| Male | 77 | 77.0 | 23 | 23.0 | 0.47 |
| Female | 186 | 80.5 | 45 | 19.5 | |
| **Highest education** | | | | | |
| No formal education | 3 | 50.0 | 3 | 50.0 | 0.37 |
| Primary | 26 | 86.7 | 4 | 13.3 | |
| Secondary | 36 | 83.7 | 7 | 16.3 | |
| Vocational | 26 | 81.3 | 6 | 18.8 | |
| Undergraduate | 102 | 76.7 | 31 | 23.3 | |
| Graduate | 71 | 80.7 | 17 | 19.3 | |

* Chi-squared test

Three of the 61 FIT-positive participants (4.9%, 95%CI = 1.0–13.7%) had advanced adenomas. A large number of participants had non-neoplastic findings, including 45.9% (28/61) with at least grade-1 hemorrhoids and 8.2% (5/61) with diverticular changes. Of those with diverticulosis, 80% (4/5) had pan-colonic disease. Twenty-five percent (24.6%, 15/61) of participants had completely normal colonoscopic examinations.

## Discussion

This study represents the first open-enrollment, single-arm, FIT-based CRC screening study of asymptomatic adults living in West Africa. Although the majority of participants had no previous experience with CRC screening, we demonstrated 85% retention for FIT analysis and endoscopic follow-up. The high rate of FIT positivity (20.5%) and low PPV for advanced neoplasia (5%) raise concerns regarding the performance of FIT for asymptomatic, average-risk CRC screening in Nigeria. However, these results must be placed in context of the exploratory nature of the study design and the small sample size. Notably, our cohort was relatively young

**Table 3. Relationship between FIT result and asymptomatic parasite infection.**

| Parasite Infection | FIT result | | | | p value |
| --- | --- | --- | --- | --- | --- |
| | Negative | | Positive | | |
| | *n* | % | *n* | % | |
| Negative | 186 | 78.8 | 50 | 21.2 | 0.05 |
| Positive | 29 | 93.5 | 2 | 6.5 | |

**Table 4. Histopathology of endoscopic findings.**

| Histopathology | *n* (12) |
|---|---|
| Adenocarcinoma | 0 |
| Traditional Adenoma | |
| Tubular | 5 |
| Tubulovillous | 1 |
| Villous | 0 |
| Serrated Adenoma | |
| Hyperplastic | 0 |
| Sessile Serrated | 0 |
| Inflammatory | 2 |
| Insufficient sample | 1 |
| Normal | 3 |
| **Location of Adenomas** | *n* (6) |
| Right colon | 4 |
| Left colon | 2 |
| Rectum | 1 |
| **Advanced Adenomas** | *n* (3) |
| Tubulovillous/villous | 1 |
| High-grade dysplasia | 0 |
| Size 1 cm | 2 |

**Table 5. Bivariate analysis for sociodemographic variables associated with adenoma detection.**

| Covariate | Adenoma detection | | | | *p* value* |
|---|---|---|---|---|---|
| | Negative | | Positive | | |
| | *n* | % | *n* | % | |
| **Age** | | | | | |
| 50 yrs | 25 | 96.2 | 1 | 3.8 | 0.23 |
| >50 yrs | 30 | 85.7 | 5 | 14.3 | |
| **Age** | | | | | |
| 40–49 yrs | 24 | 96.0 | 1 | 4.0 | 0.91 |
| 50–59 yrs | 20 | 83.3 | 4 | 16.7 | |
| 60–69 yrs | 8 | 88.9 | 1 | 11.1 | |
| 70 yrs | 3 | 100.0 | 0 | 0.0 | |
| **Monthly household income** | | | | | |
| <N49,999 ($163) | 4 | 100.0 | 0 | 0.0 | 1.00 |
| N50-99,999 ($163–327) | 5 | 100.0 | 0 | 0.0 | |
| >N100,000 ($327) | 28 | 87.5 | 4 | 12.5 | |
| **Gender** | | | | | |
| Male | 20 | 95.2 | 1 | 4.8 | 0.65 |
| Female | 35 | 87.5 | 5 | 12.5 | |
| **Highest education** | | | | | |
| No formal education | 2 | 100.0 | 0 | 0.0 | 0.89 |
| Primary | 3 | 100.0 | 0 | 0.0 | |
| Secondary | 6 | 100.0 | 0 | 0.0 | |
| Vocational | 4 | 100.0 | 0 | 0.0 | |
| Undergraduate | 26 | 89.7 | 3 | 10.3 | |
| Graduate | 14 | 82.4 | 3 | 17.6 | |

* Fischer's exact test

**Table 6. Bivariate analysis for association between personal health history and adenoma detection.**

| Covariate | Adenoma Detection | | | | p value* |
| --- | --- | --- | --- | --- | --- |
| | Negative | | Positive | | |
| | n | % | n | % | |
| **ASA or NSAID use within 7 days** | | | | | |
| No | 36 | 85.7 | 6 | 14.3 | 0.16 |
| Yes | 19 | 100.0 | 0 | 0.0 | |
| **Heart burn / dyspepsia** | | | | | |
| No | 45 | 90.0 | 5 | 10.0 | 1.00 |
| Yes | 9 | 90.0 | 1 | 10.0 | |
| **Diabetes mellitus** | | | | | |
| No | 51 | 89.5 | 6 | 10.5 | 1.00 |
| Yes | 4 | 100.0 | 0 | 0.0 | |
| **Hypertension** | | | | | |
| No | 39 | 92.9 | 3 | 7.1 | 0.36 |
| Yes | 16 | 84.2 | 3 | 15.8 | |
| **Cigarette use** | | | | | |
| No | 54 | 91.5 | 5 | 8.5 | 0.19 |
| Yes | 1 | 50.0 | 1 | 50.0 | |
| **Change bowel habits** | | | | | |
| No | 51 | 89.5 | 6 | 10.5 | 1.00 |
| Yes | 4 | 100.0 | 0 | 0.0 | |
| **Weight loss** | | | | | |
| No | 54 | 90.0 | 6 | 10.0 | 1.00 |
| Yes | 1 | 100.0 | 0 | 0.0 | |
| **Fatigue** | | | | | |
| No | 48 | 90.6 | 5 | 9.4 | 0.54 |
| Yes | 6 | 85.7 | 1 | 14.3 | |

*Fisher's Exact test

ASA–acetylsalicylic acid

NSAID–non-steroidal anti-inflammatory drug

(median age of 51 years), mostly woman (>70%), and highly educated (>60% had a university education, and thus was not representative of the general population in the region, Nigeria, or West Africa. As a consequence, the acceptability of FIT-based CRC screening should be interpreted with caution. Our findings should be viewed as laying the foundation for a larger, multi-center, population-based study with broader sociodemographic coverage.

The low PPV amongst average-risk individuals in our study may indicate a more targeted approach is needed in Nigeria. This is currently endorsed by the African Organization for Research and Training in Cancer [5]. In HICs, FIT-based screening has been validated against colonoscopy in high-risk individuals with a positive family history of CRC [24, 25]. A prospective evaluation of FIT-based screening among individuals with a known first-degree relative with CRC in Nigeria would provide valuable insight. Multi-tiered risk-stratification, similar to a model proposed by the Asian-Pacific Working Group on CRC, which included positive family history, sex, age, and cigarette use, could be developed in the Nigerian context to further enhance the performance of FIT screening [26]. More data on the natural history of CRC, including the molecular profile and unique risk factors are needed to build a similar model in Nigeria.

The low prevalence of adenomas in our study is congruent with a growing body of literature from high-volume tertiary care centers performing high-quality endoscopy across Nigeria [12–14]. A prospective, multi-center, Nigerian study of 362 patients with a recent history of self-reported rectal bleeding documented an ADR of 8.2%, despite a CRC prevalence of 18.2% (66/362) [27]. In a recent prospective study from the University College Hospital in Ibadan, Nigeria, the ADR was 9.1% among asymptomatic adults undergoing CRC screening [28]. A lower proportion of traditional adenomas may decrease FIT performance as the sensitivity of the test is inferior for non-adenomatous polyps (e.g. sessile serrated polyp) [29, 30]. The low ADR in our study also highlights the need for context-specific quality indicators for colonoscopy. An ADR of 25% in average-risk colonoscopies, which is a widely endorsed benchmark in HICs, needs to be evaluated in the context of Nigeria-specific data [31].

The demographic characteristics of our cohort may also partially explain the tests poor performance. A large proportion of participants in the study were women 50 years of age (31.0%, 103/332). Although female participants were asked to wait three days from the end of menses before providing the stool specimen for FIT analysis, this may have been an inadequate length of time and contributed to the high number of false positives. Interestingly, our study did not demonstrate an increased number of false positive FIT results with asymptomatic parasite infection. This is an important finding, as the added cost and complexity of concomitant stool collection for parasite testing is burdensome and would impact the efficacy of FIT as a population-based screening tool.

There are several limitations and potential criticisms of this study. Despite robust enrollment and a high number of positive FIT results, the number of neoplastic events was small and limits the power of the findings. The patient population represents a self-selected cohort of mostly urban volunteers with household incomes and education levels above the national average. This may have artificially enhanced the rate of participant retention, the rate of successful specimen return, and the cohorts' self-reported interest and willingness to participate in CRC screening. Culturally, our cohort was also almost exclusively from the Yoruba tribe. The characteristics of our cohort limit the generalizability of our findings and need to be validated in a larger cohort.

In conclusion, based on this exploratory analysis, CRC screening with FIT was feasible and well tolerated. However, FIT positivity (20%) was higher than expected and a high number of false positives translated into a low PPV for advanced neoplasia (4.9%) on endoscopic follow-up. Our study raises concerns about the performance of FIT for average-risk asymptomatic screening in Nigeria and need to be validated in a larger study with broader geographic and sociodemographic coverage. However, these data underscore the importance of a context-specific approach to CRC treatment in screening in West Africa.

## Supporting information

**S1 Table. Bivariate analysis for association between personal health history and FIT result.** (DOCX)

## Acknowledgments

The authors would like to acknowledge the dedicated surgical research team at OAUTHC, in particular Gbenga Samson, for their help with data entry and stool specimen processing.

## Author Contributions

**Conceptualization:** Olusegun Alatise, Olalekan Olasehinde, Funmilola Wuraola, Oluwole O. Odujoko, Olujide O. Arije, Philip E. Castle, J. Joshua Smith, Martin R. Weiser, T. Peter Kingham.

**Data curation:** Bolatito Olopade, Marguerite Samson, Olalekan Olasehinde.

**Formal analysis:** Gregory C. Knapp, Olusegun Alatise, Bolatito Olopade, Marguerite Samson, Olalekan Olasehinde, Funmilola Wuraola, Oluwole O. Odujoko, Akinwunmi O. Komolafe, Olujide O. Arije, Philip E. Castle, J. Joshua Smith.

**Investigation:** T. Peter Kingham.

**Methodology:** Gregory C. Knapp, Bolatito Olopade, Marguerite Samson, Olalekan Olasehinde, Funmilola Wuraola, Oluwole O. Odujoko, Akinwunmi O. Komolafe, Olujide O. Arije, Philip E. Castle, J. Joshua Smith, Martin R. Weiser, T. Peter Kingham.

**Project administration:** Gregory C. Knapp, Olusegun Alatise, Marguerite Samson, Olalekan Olasehinde, Funmilola Wuraola.

**Supervision:** Marguerite Samson, Philip E. Castle, J. Joshua Smith.

**Visualization:** Gregory C. Knapp.

**Writing – original draft:** Gregory C. Knapp, Olusegun Alatise, Olalekan Olasehinde, Funmilola Wuraola, Akinwunmi O. Komolafe, Olujide O. Arije, Philip E. Castle, J. Joshua Smith, T. Peter Kingham.

**Writing – review & editing:** Gregory C. Knapp, Olusegun Alatise, Bolatito Olopade, Marguerite Samson, Olalekan Olasehinde, Funmilola Wuraola, Oluwole O. Odujoko, Akinwunmi O. Komolafe, Olujide O. Arije, Philip E. Castle, J. Joshua Smith, Martin R. Weiser, T. Peter Kingham.

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
