## [Decision Letter · Decision Letter 0]

24 Nov 2020

Feasibility and performance of the fecal immunochemical test (FIT) for average-risk colorectal cancer screening in Nigeria

PONE-D-20-24828

Dear Dr. Knapp,

We’re pleased to inform you that your manuscript has been judged scientifically suitable for publication and will be formally accepted for publication once it meets all outstanding technical requirements.

Kind regards,

Sanjiv Mahadeva, MRCP, MD

Academic Editor

PLOS ONE

Journal requiremets:

1. Please provide additional details regarding participant consent. In the ethics statement in the Methods and online submission information, please ensure that you have specified (1) whether consent was informed and (2) what type you obtained (for instance, written or verbal, and if verbal, how it was documented and witnessed). If your study included minors, state whether you obtained consent from parents or guardians. If the need for consent was waived by the ethics committee, please include this information.

Additional Editor Comments (optional):

Well written manuscript with appropriate discussion.

Reviewers' comments:

Reviewer's Responses to Questions

**Comments to the Author**

1. Is the manuscript technically sound, and do the data support the conclusions?

Reviewer #1: Yes

Reviewer #2: Yes

2. Has the statistical analysis been performed appropriately and rigorously? 

Reviewer #1: Yes

Reviewer #2: Yes

3. Have the authors made all data underlying the findings in their manuscript fully available?

Reviewer #1: Yes

Reviewer #2: Yes

4. Is the manuscript presented in an intelligible fashion and written in standard English?

Reviewer #1: Yes

Reviewer #2: Yes

5. Review Comments to the Author

Reviewer #1: This study is trying to evaluate the feasibility and performance of the fecal immunochemical test FIT in Nigeria. It was well designed and clearly described, which makes the data more reliable and convincing. The disappointing part of this study is a large number of FIT-positive participants had non-neoplastic findings, a lot of whom had hemorrhoids. In this regard, the high FIT positivity and low positive predictive value for advanced neoplasia would argue its practicality and cost effectiveness in those low-income countries like Nigeria. The author should discuss more about the design of the test in the future to increase the PPV (Small sample numbers could be a key factor in the current study). Does it make sense to exclude those who had hemorrhoids for future study? On the whole, this study lays the foundation for the related study in the future.

Minor:

Define PPV (line 101)

Reviewer #2: Dear authors,

It was grateful for me to read your manuscript. I think is very interesting to know how the CRC screning could be available in Africa. Even though the sample is not representative, I consider it is important to explore the possibility to incorporate this Public Heath strategy.

I encourage you to continue this issue in order to decrease the burden of this neoplasia

6. PLOS authors have the option to publish the peer review history of their article (what does this mean?). If published, this will include your full peer review and any attached files.

Reviewer #1: No

Reviewer #2: No

---

## [Editor Report · Acceptance letter]

2 Jan 2021

PONE-D-20-24828 

Feasibility and performance of the fecal immunochemical test (FIT) for average-risk colorectal cancer screening in Nigeria 

Dear Dr. Knapp:

I'm pleased to inform you that your manuscript has been deemed suitable for publication in PLOS ONE. Congratulations! Your manuscript is now with our production department. 

Kind regards, 

on behalf of

Dr. Sanjiv Mahadeva 

Academic Editor

PLOS ONE